# Mango Consumption Is Associated with Improved Nutrient Intakes, Diet Quality, and Weight-Related Health Outcomes

**DOI:** 10.3390/nu14010059

**Published:** 2021-12-24

**Authors:** Yanni Papanikolaou, Victor L. Fulgoni

**Affiliations:** 1Nutritional Strategies, 59 Marriott Place, Paris, ON N3L 0A3, Canada; 2Nutrition Impact, 9725 D Drive North, Battle Creek, MI 49014, USA; vic3rd@aol.com

**Keywords:** NHANES, mango, nutrients, diet quality, weight-related health outcomes

## Abstract

As nutrient-dense fruits, mangoes are commonly consumed globally and are important sources of nutrients in the diet. Nonetheless, mangoes remain relatively under-consumed in the United States. The objective of the present analysis was to examine nutrient intakes, diet quality, and health outcomes using data from NHANES 2001–2018 in children and adult mango consumers (*n* = 291; adults *n* = 449) compared with mango non-consumers (children *n* = 28,257; adults *n* = 44,574). Daily energy and nutrient intakes were adjusted for a complex sample design of NHANES using appropriate weights. Mango consumption was not associated with daily energy intake, compared with non-consumption, in both children and adults. Children consuming mangoes had a significantly lower daily intake of added sugar, sodium, total fat, and a higher intake of dietary fiber, magnesium, potassium, total choline, vitamin C, and vitamin D, compared with non-consumers. In adults, mango consumers had significantly higher daily intakes of dietary fiber, magnesium, potassium, folate, vitamin A, vitamin C, and vitamin E and significantly lower intakes of added sugar and cholesterol, compared with non-consumers. Mango consumption was also associated with a better diet quality vs. mango non-consumers (*p* < 0.0001). Mango consumption in adolescents was associated with lower BMI z-scores, compared with non-consumption. In adults, BMI scores, waist circumference, and body weight were significantly lower only in male mango consumers when compared with mango non-consumers. The current results support that mango consumption is associated with improved nutrient intakes, diet quality, and certain health outcomes. Thus, dietary strategies that aim to increase mango consumption in the American population should be evaluated as part of future dietary guidance.

## 1. Introduction

Consumption of fruits in the American diet remains below authoritative dietary guidance. The 2020–2025 Dietary Guidelines for Americans (2020–2025 DGA) focuses on the inclusion of food groups that provide nutrient density while concurrently achieving caloric limits. The key recommendations comprising a healthy dietary pattern include greater consumption of vegetables, fruits, especially whole fruit, whole grains, low-fat and fat-free dairy, and lean protein foods [1]. Fruit consumption includes all fresh, frozen, dried fruits and 100% fruit juices. Data from What We Eat in America (WWEIA) 2013–2016, the dietary intake component of the National Health and Nutrition Examination Survey (NHANES), reveal that approximately 80% of the US population above the age of one are below recommendations established for fruit consumption [2]. The Centers for Disease Control and Prevention suggest that only 1 in 10 adults meet dietary guidance for fruit and vegetable consumption, placing an alarming percentage of American adults at risk for chronic diseases, including cardiovascular disease and diabetes [3]. The 2020–2025 DGA has claimed that most Americans would benefit from substantially increasing their intake of fruit, with emphasis on nutrient-dense, whole fruit. Additionally, 2020–2025 DGA suggests that increased fruit intake within a healthy dietary pattern would help increase under-consumed nutrients, including dietary fiber and potassium [1]. Dietary trends involving low carbohydrate intakes have been suggested to also be a contributory factor in the lower intake of fruit in the American population. Recent data from approximately 15,000 adults in several US communities from the Atherosclerosis Risk in Communities Study showed that both high and low percentages of carbohydrates in the diets were associated with increased mortality, with the minimal risk being 50–55% carbohydrate intake [4]. Low-carbohydrate dietary patterns have been associated with higher mortality risk in several studies [5,6,7,8]. These dietary patterns can typically be low in fruits and vegetables and have higher intakes of animal protein sources and can be higher in sodium and saturated fat content [4,8].

While dietary guidance suggests increased fruit consumption as part of healthy dietary patterns, certain fruits, including mango, remain under-consumed by Americans, even while global demand remains elevated [9]. Further, there are limited data in the published literature examining mango consumption and its association with nutrient intakes, diet quality, and health outcomes. Previous research using NHANES 2001–2008 demonstrated that mango consumption in children and adults was associated with improved nutrient and food group intakes and better diet quality, compared with those who did not consume mangoes [10]. Mangoes represent nutrient-rich fruit options, with one cup (165 g) of raw mango contributing 100 kcal, 3 g dietary fiber, 277 mg potassium, 70 µg folate, DFE, 60 mg vitamin C, and 90 µg vitamin A, RAE, 1060 µg beta-carotene and 12 mg choline [11]. Thus, the objective of the present study focused on assessing nutrient intakes, diet quality, and health outcomes using data from NHANES 2001-2018 in children and adult mango consumers with comparisons with non-consumers.

## 2. Experimental Section

The United States National Health and Nutrition Examination Survey (NHANES) is a nationally representative, cross-sectional survey of free-living, civilian residents in the US. NHANES data were collected by the National Center for Health Statistics of the Centers for Disease Control and Prevention. Detailed descriptions and analytics of NHANES have been previously documented in the scientific, peer-reviewed literature [12,13,14]. Additionally, all ethical considerations, including informed consent, were obtained for all participants or proxies, and the survey protocol was previously approved by the Research Ethics Review Board at the National Center for Health Statistics. Data from nine NHANES datasets (2001–2002; 2003–2004; 2005–2006; 2007–2008; 2009–2010; 2011–2012; 2013–2014; 2015–2016; 2017–2018) were combined for the present analysis in individuals ≥2 years of age [15,16,17,18]. Nutrient intake data for NHANES are from the relevant United States Department of Agriculture (USDA) Food and Nutrient Database for Dietary Studies (FNDDS) [19]. FNDDS are databases that provide the nutrient values for foods and beverages reported in What We Eat in America (WWEIA) [20], the dietary intake component of NHANES for each data release. 

WWEIA was collected using the automated multiple-pass method (AMPM). USDA’s AMPM represents a validated dietary data collection instrument that provides an evidence-based, efficient, and accurate format for collecting dietary intake data for large-scale national surveys [21]. The AMPM protocol is updated for each 2-year collection of WWEIA to account for the evolving food supply and address any research needs. AMPM is a fully computerized recall method that uses a 5-step interview: (1) quick list, (2) forgotten foods, (3) time and occasion, (4) detail cycle, and (5) final probe. AMPM includes an extensive compilation of standardized food-specific questions and possible options [15]. Interviewers use dietary recall status codes in both the “individual foods” and “total nutrient intakes” files to indicate the validity and reliability of responses (i.e., quality and completeness of a participant’s responses) [21]. 

### 2.1. Subjects

Sex-combined data on children (2–18 years old) and adults (19+ years old) were combined for the present analysis and differentiated as mango consumers (children *n* = 291; adults *n* = 449) or mango non-consumers (children *n* = 28,257; adults *n* = 44,574). Only data that were determined to be reliable and included completed 24 h recalled dietary data were used in the analysis. Exclusions included pregnant and lactating females and subjects presenting energy intakes equal to zero. Mango consumption was defined as participants that consumed raw mango (NHANES food code 63129010), dried mango (NHANES food code 62114050), pickled mango (NHANES food code 63129020), canned mango (NHANES food code 63129030), and frozen mango (NHANES food code 63129050). Mixed dishes containing mango were not included in the analysis. Further, the analysis used Day 1 data to define mango consumers and non-consumers, as Day 1 represented the in-person, validated data collection process.

### 2.2. Methods and Statistical Analysis

Data were analyzed using SAS software (Version 9.2, SAS Institute, Cary, NC, USA). The investigation used survey weights to develop nationally representative estimates for children and adults, followed by adjustments to consider the complex sample design of the database. The least-square means for mango consumers were compared with the least-square means for mango non-consumers, in both children and adults. Adjusted means (±standard errors) for daily intake of energy (kilocalories), nutrients, and diet quality were determined. USDA’s validated Healthy Eating Index-2015 (HEI-2015) tool was used to measure total diet quality—a measurement of alignment to authoritative dietary guidance. Energy, nutrient, and diet quality included adjustment for several variables, including age, ethnicity, sex, kilocalories (i.e., all variables with the exception of energy intake), socioeconomic status (i.e., as measured by the poverty income ratio (PIR), physical activity level, current smoking status and alcohol intake where applicable. Body mass index (BMI) was assessed in adults, and a BMI z-score was used for analyses in children. A *p*-value of ≤0.05 was deemed to represent statistical significance.

## 3. Results

### 3.1. Population Demographics

Estimated mango consumer percentages by demographic variables are presented in Table 1. No differences in mango consumption by socioeconomic status (PIR) were observed. Females had a higher consumer percentage than males. Individuals who classify as having a “vigorous physical activity” level demonstrated a significantly higher mango consumer percent vs. a sedentary lifestyle. Mexican Americans, other Hispanic, and other ethnic groups (as tracked by NHANES) had a higher mango consumer percentage, compared with White ethnicity. Current smokers had a lower consumer percentage than non-smokers.

### 3.2. Daily Nutrient and Energy Intakes: Children 2–18 Years Old

Daily nutrient and energy intake comparisons for mango consumers and non-consumers can be seen in Table 2 and Table 3. No differences were seen in energy intakes in both children and adults when comparing mango consumers and non-consumers. Mango consumption in children was associated with a significantly lower daily intake of sodium and total fat, and a higher intake of dietary fiber, magnesium, potassium, total choline, vitamin B6, vitamin C, and vitamin D, compared with mango non-consumers. In adults, mango consumers had significantly higher daily intakes of dietary fiber, magnesium, potassium, folate DFE, vitamin A, vitamin B12, vitamin C, and vitamin E and significantly lower intakes of added sugar, sodium, and cholesterol, compared with mango non-consumers.

### 3.3. Diet Quality Scores

The scores for the total and sub-categories of HEI-2015 are presented in Table 4 and Table 5. Total diet quality was significantly better in both children and adult mango consumers when compared with mango non-consumers. In children, sub-category HEI-2015 scores in mango consumers were greater for total fruit, whole fruit, and total dairy, compared with mango non-consumers. In adults, sub-category HEI-2015 scores in mango consumers were significantly higher for total fruit, whole fruit, and seafood, and plant protein, compared with mango non-consumers. Furthermore, mango consumers had lower sodium intake, compared with non-consumers. This implies that mango inclusion can be an important part of a healthy diet and likely a key contributor to overall diet quality.

### 3.4. Health Outcomes

The key health outcome examined in the present analysis related to body weight and waist circumference. In younger children, no significant differences were observed with body mass index (BMI) z-scores when comparing mango consumers and non-consumers (data not shown). BMI z-scores in older children (14–18 years old) demonstrated significant differences, which are summarized in Table 6. Indeed, adolescent mango consumers had significantly lower BMI values and waist circumferences, compared with mango non-consumers, which were primarily due to male mango consumers, rather than females. In adults, BMI, waist circumference, and body weight were significantly lower only in male mango consumers when compared with mango non-consumers.

## 4. Discussion

Our analysis of combined NHANES data shows that mangoes can be an integral part of a healthy dietary pattern. Overall, mango consumption in children was related to improved daily nutrient intakes, including higher intake of dietary fiber, magnesium, potassium, total choline, vitamin B, vitamin C, and vitamin D, and reduced intake of sodium and total fat, compared with mango non-consumers. Similarly, adult mango consumers had significantly greater daily intakes of dietary fiber, magnesium, potassium, folate DFE, vitamin A, vitamin B12, vitamin C, and vitamin E and significantly lower intakes of added sugar, sodium, and cholesterol, compared with mango non-consumers. Interestingly, dietary fiber, magnesium, potassium, folate, and vitamin A have been previously identified by authoritative dietary guidance as shortfall nutrients in the American population [22]. The current results also demonstrated that both children and adult mango consumers had a better total diet quality score when compared with mango non-consumers. Further assessment of the diet quality sub-categories in children showed that mango consumption was associated with greater intake of total fruit, whole fruit, and total dairy, compared with mango non-consumers. Similarly, in the adult population, sub-category diet quality scores in mango consumers were significantly higher for total fruit, whole fruit, and seafood and plant protein, compared with mango non-consumers. Furthermore, adult mango consumers had better sodium scores, indicative of the lower sodium intake, compared with non-consumers. The NHANES analysis also showed several key differences in health outcomes between mango consumers and non-consumers. Indeed, while no differences were seen in BMI-related analysis for all children, adolescent mango consumers had significantly lower BMI z-scores and waist circumferences, compared with mango non-consumers, which were primarily due to male mango consumers rather than females. Likewise, BMI, waist circumference, and body weight were significantly lower only in adult male mango consumers when compared with mango non-consumers.

The present findings align with previous research using data from NHANES 2001–2008 in children and adults where results demonstrated similar improvements in nutrient intakes, diet quality, and health outcomes [10]. The researchers of the previous study reported lower daily consumption of added sugars in children and adults, but also lower intakes of sodium in adults, as well as reduced body weights and decreased levels of CRP. Similar to the current study, mango consumers had better total diet quality scores, compared with non-consumers. While the previous work did not examine HEI sub-category scores, the current study examined HEI sub-category scores to determine which dietary components were leading to increased total HEI scores. Results in children from the current studyverifiedtotal fruit, whole fruit, and total dairy all significantly contributed nutrient-density [23] in the dietary pattern. Higher sub-category scores from elevated consumption of total fruit, whole fruit, seafood, and plant protein and lowered consumption of added sugar and sodium contributed to a greater total diet quality score in adult mango consumers, compared with non-consumers.

While significant scientific consensus supports that fruit represents an integral part of any dietary pattern, the American population falls alarming short of meeting recommendations [1,3,22]. Increased fruit consumption is associated with an assortment of positive health outcomes, including lowered risk of overweight and obesity, cardiovascular disease, diabetes, and cancer [24,25]. Recent research has also linked fruit and vegetable consumption with all-cause mortality. Collective analyses that included both fruit and vegetable consumption were associated with lowered risk of cardiovascular disease, cancer, and all-cause mortality, with similar findings seen when fruits were analyzed separately from vegetables. Higher consumption of apples, pears, citrus fruits, green leafy, and cruciferous vegetables was associated with lowered risk of cardiovascular disease and all-cause mortality [26]. The researchers of the previous study stated that “an estimated 5.6 to 7.8 million premature deaths globally may be attributable to a fruit and vegetable intake below 500 and 800 g/day, respectively” [26]. Thus, increased consumption of mango and mango products may help to close gaps in fruit recommendations and lower the risk of chronic disease development. Other researchers have attributed the low-carbohydrate dietary trends to further exacerbating shortfalls in fruit and vegetable consumption. Data from a prospective cohort and meta-analysis study suggested that both extremes of carbohydrate consumption—low- and high-carbohydrate diets—were associated with increased mortality risk [5]. Several previous published studies have associated low-carbohydrate diets (i.e., dietary patterns that include lower intakes of fruits, vegetables, and grains and elevated protein sources in the diet) with greater mortality risk [5,6,7,8].

The current analyses have limitations inherent in observational research and have previously been reported [27,28]. First, the results are dependent on self-reported dietary data for foods, which may involve study participants under- or over-estimating food consumption, leading to inaccuracies in energy and nutrient intakes. Second, data were gathered using a 24 h dietary recall, which relies on the memory of study participants/caregivers, and while validated methods were used to collect data, recall information was subject to inaccuracies and bias from memory challenges and other potential measurement errors experienced in epidemiological investigations [29,30]. Third, our current analysis considered dietary patterns with and without mango consumption; hence, other food choices within an individual’s eating pattern may also contribute to relationships observed with nutrient intakes. For example, our data indicate higher HEI sub-component scores (i.e., category 3 and 4 of the HEI-2015 scale) for total fruit and whole fruit in both children and adults; thus, it is probable to suggest that mango consumers are more likely to consume greater amounts of fruits in their diet, leading to improved nutrient intakes and diet quality. Based on our findings, it is recommended that future research on the American diet identify fruit and vegetable dietary patterns and associations with nutrient intakes, diet quality, and various health outcomes.

A significant benefit of using NHANES data for the current analyses includes access to a large and nationally representative dataset of adults of various age groups in the US and corresponding food and nutrient intake data. As the present research was observational, and since growth and development are multifactorial, future research designs will need to consider randomized, controlled trials.

## 5. Conclusions

Our analysis demonstrated several associations between mango consumers, nutrient intakes, diet quality, and weight-related health outcomes. Mango consumption in children was associated with a higher intake of dietary fiber, magnesium, potassium, total choline, vitamins B6, C, and D, and reduced intake of sodium and total fat, compared with mango non-consumers. Adults including mangoes in their diet had significantly greater daily intakes of dietary fiber, magnesium, potassium, folate DFE, vitamins A, B12, C, and E, and significantly lower intakes of added sugar, sodium, and cholesterol, compared with mango non-consumers. Mango consumers also demonstrated a better overall diet quality when compared with mango non-consumers. Weight-related health outcome assessment showed that adolescent mango consumers had significantly lower BMI z-scores and waist circumferences, compared with mango non-consumers, which were primarily due to male mango consumers rather than females. Likewise, BMI, waist circumference, and body weight were significantly lower only in adult male mango consumers when compared with mango non-consumers. The present findings are aligned with previously published data documenting numerous benefits associated with the inclusion of fruit within healthy dietary patterns.

## Figures and Tables

**Table 1 nutrients-14-00059-t001:** Estimated percentage of mango consumers by levels of demographic variables.

Variable	% Mango Consumers	SE	LCL	UCL	*p*
Age 2–3	1.229	0.338	0.562	1.897	
Age 4–8	1.160	0.186	0.792	1.528	0.854
Age 9–13	0.627	0.113	0.403	0.851	0.084
Age 14–18	0.458	0.100	0.260	0.655	0.026
Age 19–50	0.841	0.095	0.654	1.028	0.269
Age 51–70	0.713	0.092	0.531	0.895	0.137
Age 71+	0.639	0.127	0.388	0.889	0.096
Sex = Male	0.660	0.055	0.552	0.769	
Sex = Female	0.916	0.088	0.743	1.090	0.003
PA = Sedentary	0.609	0.095	0.422	0.797	
PA = Moderate	0.745	0.085	0.577	0.913	0.221
PA = Vigorous	0.915	0.090	0.736	1.093	0.013
PIR < 1.35	0.747	0.081	0.587	0.907	
1.35 <= PIR <= 1.85	0.821	0.145	0.534	1.108	0.629
PIR > 1.85	0.737	0.071	0.597	0.877	0.922
Ethnicity = Mexican American	2.015	0.203	1.613	2.417	<0.0001
Ethnicity = Other Hispanic	1.464	0.263	0.944	1.985	0.0002
Ethnicity = White	0.412	0.056	0.302	0.523	
Ethnicity = Black	0.519	0.087	0.346	0.692	0.263
Ethnicity = Other	2.459	0.330	1.805	3.112	<0.0001
Smoking Current = No	0.878	0.069	0.742	1.013	
Smoking Current = Yes	0.334	0.095	0.147	0.521	<0.0001

SE = standard error; LCL = lower confidence level; UCL = upper confidence level; PA = physical activity; PIR = poverty income ratio (a measure of socioeconomic status).

**Table 2 nutrients-14-00059-t002:** Day 1 nutrient and energy intakes in mango non-consumers vs. mango consumers: children aged 2–18 years old.

Energy/Nutrients	Mango Non-Consumers		Mango Consumers		*p*
	Mean	SE	Mean	SE	
Energy (kcal)	1888	10	1990	65	0.113
Carbohydrate (g)	249	0.7	257	4.6	0.076
Added sugars (tsp eq)	17.3	0.2	16.0	0.9	0.162
Total sugars (g)	117	0.7	134	4.0	0.0001
Protein (g)	67	0.4	71	3.6	0.313
Total fat (g)	71	0.3	67.4	1.3	0.002
Total MUFA (g)	24.2	0.1	22.4	0.6	0.002
Total PUFA (g)	15.8	0.1	14.6	0.6	0.050
Total SFA (g)	24.9	0.1	24.1	1.0	0.372
Cholesterol (mg)	216	2.1	244	21.6	0.200
Dietary fiber (g)	13.9	0.1	17.0	0.8	0.0002
Calcium (mg)	1020	8.0	1101	42.0	0.066
Folate, DFE (µg)	504	5.5	559	36.0	0.131
Iron (mg)	13.7	0.1	13.6	0.6	0.763
Lutein + zeaxanthin (µg)	795	18.9	884	111	0.430
Magnesium (mg)	234	1.1	259	7.3	0.001
Niacin (mg)	21.2	0.2	20.8	1.2	0.718
Phosphorus (mg)	1263	6.0	1345	34.1	0.022
Potassium (mg)	2155	11.1	2521	70.9	<0.0001
Riboflavin (Vitamin B2) (mg)	1.9	0.01	2.1	0.1	0.038
Sodium (mg)	2995	14.8	2720	90.5	0.003
Thiamin (Vitamin B1) (mg)	1.5	0.01	1.5	0.06	0.984
Total choline (mg)	249	1.6	294	17.1	0.010
Vitamin A, RAE (µg)	592	6.0	646	29.9	0.077
Vitamin B12 (µg)	4.7	0.1	4.7	0.3	0.777
Vitamin B6 (mg)	1.7	0.02	1.9	0.1	0.040
Vitamin C (mg)	74	1.5	124	8.7	<0.0001
Vitamin D (µg)	5.5	0.1	6.8	0.5	0.016
Vitamin E (mg)	7.0	0.1	7.5	0.3	0.064
Zinc (mg)	9.9	0.1	10.5	0.7	0.330

Mean = least-square mean; SE = standard error; MUFA = monounsaturated fatty acids; PUFA = polyunsaturated fatty acids; SFA = saturated fatty acids; vitamin D = D_2_ + D_3_; vitamin E = as α-tocopherol; DFE = dietary folate equivalents. NHANES 2001–2018.

**Table 3 nutrients-14-00059-t003:** Day 1 nutrient and energy intakes in mango non-consumers vs. mango consumers: adults aged 19+ years old.

Energy/Nutrients	Mango Non-Consumers		Mango Consumers		*p*
	Mean	SE	Mean	SE	
Energy (kcal)	2153	6.6	2194	53.2	0.452
Carbohydrate (g)	254	0.6	266	4.1	0.004
Added sugars (tsp eq)	17.5	0.2	14.9	0.8	0.001
Total sugars (g)	112	0.5	122	3.9	0.010
Protein (g)	83	0.3	80	3.3	0.334
Total fat (g)	84	0.2	82	2.0	0.285
Total MUFA (g)	29.5	0.1	29.5	1.1	0.988
Total PUFA (g)	19.5	0.1	18.7	0.6	0.155
Total SFA (g)	27.2	0.1	26.2	0.9	0.229
Cholesterol (mg)	293	1.9	259	16.4	0.042
Dietary fiber (g)	17.2	0.1	22.2	0.6	<0.0001
Calcium (mg)	982	5.0	1017	46	0.445
Folate, DFE (µg)	531	3.4	629	27.2	0.001
Iron (mg)	14.7	0.1	15.6	0.7	0.219
Lutein + zeaxanthin (µg)	1649	48.3	2096	460	0.330
Magnesium (mg)	307	1.5	347	10.7	0.001
Niacin (mg)	26.4	0.1	25.3	0.6	0.067
Phosphorus (mg)	1409	4.7	1423	60.7	0.816
Potassium (mg)	2696	12.0	2970	59.7	<0.0001
Riboflavin (Vitamin B2) (mg)	2.2	0.01	2.3	0.1	0.275
Sodium (mg)	3577	9.0	3246	106	0.003
Thiamin (Vitamin B1) (mg)	1.6	0.01	1.63	0.06	0.879
Total choline (mg)	338	1.4	330	15.6	0.628
Vitamin A, RAE (µg)	644	9.1	746	50.5	0.050
Vitamin B12 (µg)	5.2	0.1	4.7	0.2	0.018
Vitamin B6 (mg)	2.2	0.02	2.3	0.1	0.138
Vitamin C (mg)	80	1.2	141	7.5	<0.0001
Vitamin D (µg)	4.7	0.1	5.0	0.5	0.561
Vitamin E (mg)	9.1	0.1	11.2	0.5	0.0002
Zinc (mg)	11.4	0.1	11.2	0.4	0.628

Mean = least-square mean; SE = standard error; MUFA = monounsaturated fatty acids; PUFA = polyunsaturated fatty acids; SFA = saturated fatty acids; vitamin D = D_2_ + D_3_; vitamin E = as α-tocopherol; DFE = dietary folate equivalents. NHANES 2001–2018.

**Table 4 nutrients-14-00059-t004:** Day 1 Healthy Eating Index (HEI)-2015 and sub-category mean scores in children.

HEI Total and 12 HEI Sub-Categories	Mango Non-Consumers		Mango Consumers		*p*
	Mean	SE	Mean	SE	
Total vegetables (Category 1)	2.12	0.02	2.04	0.2	0.678
Greens and beans (Category 2)	0.93	0.03	1.26	0.2	0.153
Total fruit (Category 3)	2.53	0.04	4.06	0.1	<0.0001
Whole fruit (Category 4)	2.34	0.04	4.41	0.1	<0.0001
Whole grains (Category 5)	2.68	0.05	3.36	0.5	0.180
Total dairy (Category 6)	6.91	0.05	7.49	0.3	0.024
Total protein foods (Category 7)	3.58	0.03	3.67	0.2	0.559
Seafood and plant protein (Category 8)	1.62	0.03	1.99	0.3	0.237
Fatty acid ratio (Category 9)	3.91	0.05	3.50	0.4	0.294
Sodium (Category 10)	4.92	0.06	5.85	0.5	0.056
Refined grains (Category 11)	5.09	0.05	5.60	0.4	0.183
Saturated fat (Category 12)	5.43	0.05	5.67	0.5	0.642
Added sugar (Category 13)	6.03	0.05	6.48	0.3	0.135
Total	48.10	0.20	55.38	1.7	<0.0001

Mean = least-square mean; SE = standard error; NHANES 2001–2018.

**Table 5 nutrients-14-00059-t005:** Day 1 Healthy Eating Index (HEI)-2015 and sub-category mean scores in adults.

HEI Total and 12 HEI Sub-Categories	Mango Non-Consumers		Mango Consumers		*p*
	Mean	SE	Mean	SE	
Total vegetables (Category 1)	3.06	0.02	3.29	0.1	0.075
Greens and beans (Category 2)	1.54	0.03	1.92	0.2	0.067
Total fruit (Category 3)	2.01	0.03	3.99	0.1	<0.0001
Whole fruit (Category 4)	2.05	0.03	4.44	0.1	<0.0001
Whole grains (Category 5)	2.55	0.04	3.13	0.3	0.098
Total dairy (Category 6)	5.07	0.03	5.29	0.3	0.413
Total protein foods (Category 7)	4.22	0.01	4.03	0.1	0.184
Seafood and plant protein (Category 8)	2.34	0.03	2.82	0.2	0.024
Fatty acid ratio (Category 9)	5.03	0.04	5.36	0.4	0.343
Sodium (Category 10)	4.25	0.03	5.57	0.4	0.001
Refined grains (Category 11)	6.22	0.04	6.47	0.4	0.516
Saturated fat (Category 12)	5.88	0.04	6.15	0.3	0.391
Added sugar (Category 13)	6.73	0.04	7.46	0.3	0.007
Total	50.95	0.21	59.94	1.3	<0.0001

Mean = least-square mean; SE = standard error; NHANES 2001–2018.

**Table 6 nutrients-14-00059-t006:** Weight-related health outcomes in mango consumers and non-consumers, for both children and adults.

Health Outcome	Sex	Mango Non-Consumers		Mango Consumers		*p*
		Mean	SE	Mean	SE	
Children, 14–18 years old						
BMI z-score	All	1.2	0.02	0.9	0.2	0.166
	Male	1.2	0.03	0.7	0.1	0.0001
	Female	1.1	0.03	1.4	0.3	0.327
Waist circumference (cm)	All	83.5	0.4	77.0	2.7	0.019
	Male	84.2	0.6	77.0	2.3	0.002
	Female	82.8	0.5	77.5	4.3	0.236
Body weight (kg)	All	69.5	0.5	62.9	3.4	0.060
	Male	73.6	0.7	67.2	4.5	0.151
	Female	65.2	0.6	59.8	4.8	0.263
Adults, 19+ Years Old						
BMI	All	29.2	0.1	28.9	0.9	0.766
	Male	29.0	0.1	26.9	0.6	0.001
	Female	29.4	0.1	30.3	1.4	0.516
Waist circumference (cm)	All	99.5	0.2	97.9	2.1	0.433
	Male	101.8	0.3	97.0	1.3	0.0004
	Female	97.2	0.3	97.7	3.3	0.888
Body weight (kg)	All	83.2	0.3	82.6	2.8	0.830
	Male	89.7	0.4	83.8	1.9	0.003
	Female	76.9	0.3	79.8	4.3	0.509

Mean = least-square mean; SE = standard error; BMI = body mass index; NHANES 2001–2018.

## Data Availability

Publicly available datasets were analyzed in this study. This data can be found here: https://wwwn.cdc.gov/nchs/nhanes (accessed on 19 November 2021).

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
