# Peer review of "Mango Consumption Is Associated with Improved Nutrient Intakes, Diet Quality, and Weight-Related Health Outcomes"

_nutrients, 2021, doi:10.3390/nu14010059_

Round 1

Reviewer 1 Report

Overall, very interesting paper.  I have a few comments to strengthen the paper.

  1. Use “sex” rather than “gender”
  2. Describe how you analyzed sex differences.
  3. All tables need to be better positioned (right justified and in line with text or the entire width.
  4. Use 3 decimal places for p-levels.
  5. Standard Deviations and 95% CI’s should be reported on all tables.
  6. The percentage of the population consuming mango’s is less than 1% in most parameters yet you are attributing significant health benefits.  Discuss how this low prevalence could impact on health outcomes. 
  7. Discuss whether people who consume mango’s may be more apt to consume fruits in their diet and whether that may have an impact of overall results.  Would tracking other fruits reveal similar findings?  You address this in the discussion somewhat but I don’t think you make the point strong enough.
  8. You indicate “…increased consumption of mango and mango products may help to close gaps in fruit recommendations and lower the risk of chronic disease development”.  Maybe I missed it but do you have data showing that those who consume similar amounts of fruits but not mango’s have different effects?  Where the overall fruit intake similar?  It seems your data only looks as mango vs. non-mango consumers.  I think you address this in the limitation but some additional discussion above that may help delineate.

Author Response

Dear Reviewer,

Thank you for taking the time to review our paper and provide feedback. We have provided answers and comments to your questions and suggestions below. Please let us know if you have any further questions or suggestions. The authors’ answers are in bold font.

Sincerely,

Yanni Papanikolaou and Victor L. Fulgoni

Reviewer’s Question/Comment: Use ‘sex’ rather than ‘gender’.

Authors’ Response: Thank you for this feedback. The revision has been made to use ‘sex’ vs. ‘gender’.

Reviewer’s Question/Comment: Describe how you analyzed sex differences.

Authors’ Response: Thank you for following up with this question. Using NHANES data, subjects were considered mango consumers if they ate a mango foodcode on day 1 of dietary data collection. Day 1 dietary weights were used in all regression analyses. The least square means for mango consumers was compared to the least square means for non-consumers of mango. The analysis also used several covariates, including age, sex, ethnicity, poverty income ratio level, physical activity level, current smoking status and alcohol intake. For the current analysis, we only considered differences between male mango consumers and non-consumers and used a similar approach for females. Please let us know if you have further questions or comments.

Reviewer’s Question/Comment: Use 3 decimal places for p-levels.

Authors’ Response: We have revised our manuscript to only use 3 decimal places for p-levels, except for p<0.0001 and scenarios where the p-level may be 0.0002, 0.0001 etc. Thank you.

Reviewer’s Question/Comment: Standard deviations and 95% CI’s should be reported on all tables.

Authors’ Response: Thank you for your comment. We have chosen to report standard errors vs. standard deviations and CI’s since we were focused on reporting on the accuracy of the means in addition to comparing mean differences between mango consumers and non-consumers. We have used this statistical approach on all of our previous, peer-reviewed publications.

Reviewer’s Question/Comment: The percentage of the population consuming mango’s is less than 1% in most parameters yet you are attributing significant health benefits. Discuss how this low prevalence could impact on health outcomes.

Authors’ Response: You are correct in that our study identified a small percentage of the population that consumes mangoes. Our study also shows that mangoes, as part of the current dietary pattern (as described by the NHANES data collected) is associated with significantly greater nutrient intakes, improved diet quality and weight-related health outcome benefits. Our study not only considers mango intake, but rather the whole day dietary intake pattern. We also propose that mango consumers tend to have healthier diets overall, including greater consumption of fruits and whole grains—key attributes that are likely contributing to the results we observed.

Reviewer’s Question/Comment: Discuss whether people who consume mangoes may be more apt to consume fruits in their diet and whether that may have an impact on overall results. Would tracking other fruits reveal similar findings? You address this in the discussion somewhat, but I don’t think you make the points strong enough.

Authors’ Response: Yes, we believe that mango consumers are more likely to consume greater amounts of fruits in their diet and this would play a role in the overall results. Future research can identify existing fruit dietary patterns and associations with nutrient intakes, diet quality and various health outcomes. We have expanded the discussion section as recommended. The second last paragraph of the discussion reads as follows:

Our current analysis considered dietary patterns with and without mango consumption, hence, other food choices within an individual’s eating pattern may also contribute to relationships observed with nutrient intakes. For example, our data indicate higher HEI sub-component scores (i.e., category 3 and 4 of the HEI-2015 scale) for total fruit and whole fruit in both children and adults, thus, it is probable to suggest that mango consumers are more likely to consume greater amounts of fruits in their diet, leading to improved nutrient intakes and diet quality. Based on our findings, it is recommended that future research in Americans identify fruit dietary patterns and associations with nutrient intakes, diet quality and various health outcomes.

Reviewer’s Question/Comment: You indicate “…increased consumption of mango and mango products may help close gaps in fruit recommendations and lower the risk of chronic disease development.” Maybe I missed it, but do you have data showing that those who consume similar amounts of fruits, but not mangoes, have different effects? Were the overall fruit intakes similar? It seems your data only looks at mango vs. non-mango consumers. I think you address this in the limitations, but some additional discussion above may help delineate.

Authors’ Response: The current analysis did not consider consumption of other types of fruit. However, we are quite interested in examining fruit dietary patterns in the near future, as we believe this will be an important addition in the peer-reviewed, published literature. Please refer to the additions made above in reference to the limitations portion of the discussion section.

Reviewer 2 Report

In my opinion, the manuscript was prepared well.

  • The Introduction Section explains the design of the study. The Authors well justify the research topic.
  • The Descriptions of the results were correct.
  • The presented tables were prepared precisely and also legible.
  • The Conclusions were well formulated.

Author Response

Dear Reviewer,

Thank you for your feedback. It is great appreciated. If you have any further comments, please feel free to let us know.

Regards,

Yanni Papanikolaou, Victor L. Fulgoni, III